# Transcriptomics of Long-Term Meditation Practice: Evidence for Prevention or Reversal of Stress Effects Harmful to Health

**DOI:** 10.3390/medicina57030218

**Published:** 2021-03-01

**Authors:** Supaya Wenuganen, Kenneth G. Walton, Shilpa Katta, Clifton L. Dalgard, Gauthaman Sukumar, Joshua Starr, Frederick T. Travis, Robert Keith Wallace, Paul Morehead, Nancy K. Lonsdorf, Meera Srivastava, John Fagan

**Affiliations:** 1Department of Physiology and Health, Maharishi International University, Fairfield, IA 52556, USA; wenuganen@miu.edu (S.W.); kwalton@miu.edu (K.G.W.); kwallace@miu.edu (R.K.W.); pmorehead@miu.edu (P.M.); healthoffice@drlonsdorf.com (N.K.L.); 2Institute for Prevention Research, Fairfield, IA 52556, USA; 3Cancer Genomics Research Laboratory (CGR), Division of Cancer Epidemiology and Genetics, NCI Leidos Biomedical Research, Inc., Gaithersburg, MD 20877, USA; shilpareddy.20k@gmail.com; 4Department of Anatomy, Physiology, and Genetics, Uniformed Services University of the Health Sciences, Bethesda, MD 20814, USA; clifton.dalgard@usuhs.edu (C.L.D.); gauthaman.sukumar.ctr@usuhs.edu (G.S.); joshua.starr.ctr@usuhs.edu (J.S.); 5Center for Brain, Cognition, and Consciousness, Maharishi International University, Fairfield, IA 52557, USA; ftravis@miu.edu; 6Health Research Institute, Fairfield, IA 52556, USA

**Keywords:** chronic stress, transcendental meditation, gene expression, energy metabolism, biological aging, epigenetic effects, allostatic load

## Abstract

*Background and Objectives***:** Stress can overload adaptive mechanisms, leading to epigenetic effects harmful to health. Research on the reversal of these effects is in its infancy. Early results suggest some meditation techniques have health benefits that grow with repeated practice. This study focused on possible transcriptomic effects of 38 years of twice-daily Transcendental Meditation^®^ (TM^®^) practice. *Materials and Methods:* First, using Illumina^®^ BeadChip microarray technology, differences in global gene expression in peripheral blood mononuclear cells (PBMCs) were sought between healthy practitioners and tightly matched controls (*n* = 12, age 65). Second, these microarray results were verified on a subset of genes using quantitative polymerase chain reaction (qPCR) and were validated using qPCR in larger TM and control groups (*n* = 45, age 63). Bioinformatics investigation employed Ingenuity^®^ Pathway Analysis (IPA^®^), DAVID, Genomatix, and R packages. *Results:* The 200 genes and loci found to meet strict criteria for differential expression in the microarray experiment showed contrasting patterns of expression that distinguished the two groups. Differential expression relating to immune function and energy efficiency were most apparent. In the TM group, relative to the control, all 49 genes associated with inflammation were downregulated, while genes associated with antiviral and antibody components of the defense response were upregulated. The largest expression differences were shown by six genes related to erythrocyte function that appeared to reflect a condition of lower energy efficiency in the control group. Results supporting these gene expression differences were obtained with qPCR-measured expression both in the well-matched microarray groups and in the larger, less well-matched groups. *Conclusions:* These findings are consistent with predictions based on results from earlier randomized trials of meditation and may provide evidence for stress-related molecular mechanisms underlying reductions in anxiety, post-traumatic stress disorder (PTSD), cardiovascular disease (CVD), and other chronic disorders and diseases.

## 1. Introduction

McEwen and Akil [1] recently outlined major steps of progress from 50 years of research on the neurobiology of stress. The concept of stress responses as adaptive mechanisms capable of being overworked or overloaded, thereby producing an “allostatic load” that increases propensity for disease, is one of many contributions from the McEwen laboratory [1,2]. Repeated exposure to even mild stressors can constrain adaptive mechanisms, contributing to cardiometabolic, cognitive, and behavioral disorders [1,2,3]. Such approaches to modeling physiological dysregulation predict not only illness but also longevity [4,5]. Moreover, psychosocial stress is associated with elevated oxidative stress [6,7], a further contributor to disease and aging. The high prevalence of stress-related diseases naturally motivates the search for effective interventions.

Accumulated research on the standardized Transcendental Meditation^®^ (TM^®^) technique suggests it can reduce unwanted effects of stress. This technique was revived from the ancient Vedic tradition and made available to the world at large by Maharishi Mahesh Yogi starting in the 1950s [8,9]. The first research on physiological effects appeared in 1970 [10]. As reported and reviewed by others, investigations of the TM program as a therapeutic intervention have reported benefits for PTSD [11,12,13], anxiety disorders [14], and risk factors for cardiovascular disease (CVD) [15,16,17,18]. Moreover, studies on healthcare utilization suggest reductions in a wide spectrum of diseases [19,20]. Evidence supports the conclusion that the automatic self-transcending nature of this technique bears primary responsibility for these effects [21,22].

Research on long-term molecular and cellular effects of stress that relate to disease and aging has focused on the immune system and mitochondrial energy production. Early investigations found differences affecting glucocorticoid and inflammatory signaling [23]. Further research led to the discovery of a “Conserved Transcriptional Response to Adversity (CTRA)” accompanying severe or chronic stress [24,25]. The CTRA involves upregulation of pro-inflammatory genes and downregulation of antiviral and antibody components of the defense response [26].

Energy metabolism also is a key component of stress responses and adaptation [27], and mitochondrial energy production is affected by stress [28,29]. Both chronic stress and aging are associated with reductions in mitochondrial function and energy efficiency [30,31].

Some mind–body interventions have been reported to decrease or reverse such effects of stress [32,33,34], including effects on the “epigenetic clock”, a reproducible biomarker of biological aging [35]. However, meditation studies and other mind–body intervention research, especially studies examining transcriptomic effects, have tended to be of short duration [36,37]. This leaves a gap in our knowledge of effects deriving from decades-long practice of these programs. Based on prior research indicating that effects of the TM program are cumulative, we hypothesized that long-term practice would produce transcriptomic differences connected to the program’s stress-reducing and health-promoting benefits. Results of this study appear to support that hypothesis. Due to budgetary constraints and to this being the first of its kind, the study is exploratory, not definitive, in nature. Nevertheless, it employs the three steps characterizing gene expression comparisons, namely a discovery step (microarray comparison [38]), a verification and validation step applying a quantitatively more accurate approach (qPCR) to a sample of genes, and a functional analysis showing consistency with prior research on stress effects and meditation. The results appear to provide promising avenues for further investigation.

## 2. Materials and Methods

### 2.1. Research Design and Participants

This study used DNA microarray transcriptomics and qPCR technologies in peripheral blood mononuclear cells (PBMCs) to compare non-practitioner control groups and TM practitioner groups. First, two demographically well-matched small groups were selected from a larger pool of volunteers for the microarray analysis (Table 1). Later, qPCR analysis of 15 genes selected from those differentially expressed in the microarray was used to test the reproducibility of the microarray results in the larger pool of 45 volunteers. All study participants were recruited through advertising on the campus of Maharishi International University and in public places in or near Fairfield, Iowa. The design and methods were approved by the University’s Institutional Review Board. Following both written and oral descriptions of the study, participants gave signed consent prior to participation.

To reduce genetic variation, study participants were limited to self-identified white males and females. Prospective participants were excluded if they reported a doctor-identified history of diabetes, nerve damage, heart attack, coronary heart disease, stroke, kidney failure, cancer, any other life-threatening illness, a major psychiatric disorder, or substance abuse. In addition, candidates for the control group were excluded if they had ever been instructed in the TM program. Practitioners of the TM program were excluded if they had not regularly practiced the program twice a day or usually twice a day.

The meditation program consisted of the standard TM technique [9] practiced in the microarray TM group for 458 ± 49 months, with later addition of the TM-Sidhi^®^ program, also practiced twice daily in this group, for 406 ± 50 months. The TM-Sidhi program, like the TM program, is drawn from the ancient Vedic tradition and is said to promote more rapid incorporation of the benefits of TM practice into daily life [9]. As with the small groups for microarray analysis, the larger groups used for qPCR analysis were not significantly different in age (Table 2). The TM group was notably higher in the number of vegetarians and tended to be higher in other indicators of healthy lifestyle. Subjective socioeconomic status (SES) data were not available. TM in this qPCR study was practiced for 475 ± 40 months, and the TM-Sidhi program for 396 ± 44 months. 

### 2.2. PBMC Preparation

Blood was drawn in random order from 4–6 participants a day between 10 AM and 4 PM by a certified phlebotomist using a 19-gauge butterfly needle. A total of 16 mL was drawn in two collection tubes (BD Vacutainer^®^ CPT^TM^ Mononuclear Cell Preparation Tube). The buffy coat containing the PBMCs was harvested according to the manufacturer’s instructions, and the cell pellet was stored at −80 °C.

### 2.3. RNA Extraction, Concentration Measurement, and Integrity Check

RNA was extracted from the buffy coat employing the RNAzol B Kit (Ambion^®^). RNA concentration was estimated using a UV absorption ratio method [39]. Ratios above 1.7 were considered sufficiently pure for the sample to be used for microarray and qPCR analyses. RNA integrity was analyzed by automated electrophoresis on microfluidic labchips using the Experion^TM^ System (Bio-Rad). RNA Quality Indicator (RQI) values considered acceptable were in the range 7 < RQI ≤ 10. Samples meeting the criterion for integrity were selected for array-based expression profiling. 

### 2.4. Whole-Genome mRNA Expression Using Bead-Based Array

Total RNA samples from matched TM and control groups were shipped on dry ice to the Genomics Core at the University of Chicago, Chicago, IL, for genome-wide mRNA expression profiling using Illumina^®^ Gene Expression BeadChip technology. A minimum amount of 50 ng of total RNA for each sample was labeled using the Illumina TotalPrep^TM^ RNA Labeling Kit (Ambion). Labeled samples were incubated on Illumina HumanHT-12v4 Expression BeadChips containing probes for 47,231 features and imaged using the Illumina iScan system. Raw data from the iScan were converted using the GenomeStudio software package and Gene Expression Module (Illumina), resulting in identification of 16,247 genes and loci.

Microarray data were subjected to statistical analysis using R BioConductor (lumi and BeadArray) packages that include methods correcting for Type 1 and Type 2 errors [40]. Differential expression for each individual gene was considered acceptable if it met comparatively strict criteria, i.e., ratio of TM and control group gene-expression values (normalized by the quantile method) ≥2.0 and *p*-value for differential expression ≤0.05. A heat map with hierarchical clustering was created from data on the differentially expressed genes using the average linkage method and the Pearson distance metric to show how data aggregated. This analysis was performed in R version 2.11.1 with the gplots package using hclust and heapmap.2 functions [41]. Networks with molecular paths plausibly affected by practice of the TM program were identified using Ingenuity^®^ Pathway Analysis (IPA^®^) software [42].

To assign functional annotations for differentially expressed genes, gene ontological enrichment analysis was performed through the DAVID database, a free online web-based tool. Differentially expressed genes submitted to DAVID were sorted into lists of ranked genes under enriched gene ontological process terms. The *p*-values displayed are a DAVID adjustment of the Fisher exact test *p*-value from testing for a significantly higher number of genes in the submitted list belonging to the group of genes, compared with all genes in the human genome [43]. 

A similar approach was used in the disease association analysis, in this case using Genomatix. Genomatix is data-mining software for extracting and analyzing gene relationships from literature databases such as NCBI PubMed and annotation data such as Gene Ontology. The software calculates overrepresentation of specific biological terms within the input and ranks genes related to specific diseases in the output. The program calculates *p*-values in a similar manner to that described for the DAVID online tool.

### 2.5. qPCR Analysis

Target genes chosen for validation of the microarray results included genes representing the main findings of the microarray component, including key relationships to health and aging. The primers were designed using Primer BLAST and were purchased from IDT^®^ Technologies (http://www.idtdna.com/site, accessed 5 January 2014). The target genes, the reference gene, and their primer sequences are shown in Appendix A: Primer Sequences for qPCR Reactions. 

First, for each participant, complementary DNA (cDNA) was constructed from the extracted RNA employing the iScript^TM^ cDNA Synthesis Kit (Bio-Rad). Each cDNA reaction mixture of 20 µL contained 1 µg of the RNA template and 10 µL of master mix (containing 1.0 µL iScript reverse transcriptase solution, 4.0 µL of 5× iScript mix, and 5 µL nuclease-free water). The PCR reaction was run in the GeneAmp^®^ PCR 9700 systen (Applied Biosystems) in the following steps: 5 min at 25 °C; 30 min at 42 °C; 3.5 min at 85 °C, and hold at 4 °C.

Second, the qPCR experiment was conducted using the SsoFast^TM^ EvaGreen^®^ Supermix (Bio-Rad). Each reaction contained 5 µL 1× SsoFast EvaGreen supermix, a final concentration of 320 nM each of forward primer and reverse primer, 5 ng cDNA template, and 2.9 µL nuclease-free water in a 10-µL final volume. Finally, samples were assayed in quadruplicate in a 320-well reaction plate using a C1000 thermal cycler (Bio-Rad) combined with a CFX 384 Real-Time System (Bio-Rad) in the following cycling steps: 1. Enzyme activation: 95 °C, 30 s, 1 cycle; 2. Denaturation: 95 °C, 3 s, 40 cycles; 3. Annealing/Extension: 58 °C, 5 s, 40 cycles; 4. Melt curve: 65–95 °C (in 0.5 °C increments, 5 s/step), 1 cycle.

The relative expression level of each gene (“fold difference”) was calculated using the (2.0 ^(∆∆Ct)^) method [44]. Statistical analysis was performed using ANOVA in SPSS. An alpha level of *p* ≤ 0.05 was adopted for statistical significance.

## 3. Results

### 3.1. Differential Expression, Heat Map, and Clustering Analysis from the Microarray Study

Gene expression profiles for the 200 genes and loci that met the cut-off criteria are shown in Figure 1. The majority of these (136) were downregulated in the TM group. Numerical gene expression data, including expression ratios and *p*-values for all 200 genes and loci, are available in Appendix A: Differentially Expressed Genes and Loci. The hierarchical clustering analysis based on Pearson correlation distances showed distinct expression patterns for each group. One gene, *SLC6A4*, coding for the serotonin reuptake transporter, whose *p*-value exceeded the chosen ≤0.05 cut-off by a small margin (actual *p* = 0.057), was included because of its known importance in stress-related mechanisms and health. The expression ratios of all genes and loci met the criterion of ≥2.0.

### 3.2. qPCR Validation of Microarray Data

Verification and validation of the microarray data were conducted on 15 key genes selected from the 200. Verification of microarray data on these selected genes compared microarray output to results obtained with qPCR (Table 3). Preliminary validation (Table 4) involved larger groups that were less well-matched demographically (see Table 2) than those in the microarray comparison (see Table 1).

### 3.3. Top Networks

Principal networks that reflect possible causal and functional significance were found by applying IPA to the 200 genes. The network with the highest p-score, along with its top functions and diseases, is shown in Figure 2. The network with the second highest p-score is shown in Figure 3 (Additional networks with lower p-scores are shown in Appendix A: Networks 3–5). Each network includes molecules central to the network (core molecules) and other molecules that affect or are affected by the core molecules. Network 1 (Figure 2), with a p-score of 37 (i.e., *p* = 1 × 10^−37^), is strongly related to stress, inflammation, and the defense response. The largest number of connections are to the NF-κB complex and the interferon alpha complex.

Three differentially expressed genes in Network 1—*IL1B* and *TLR4* (upregulated in the TM group) and *SOCS3* (downregulated in the TM group)—are connected not only to the NF-κB complex but also to interferons alpha and gamma, immunoglobulin, and other core molecules. The major themes of this network are cell-to-cell signaling and interaction, hematologic system development and function, and inflammatory response. Key canonical pathways (CPs) are highlighted in the figure. 

Core molecules of Network 2 (p-score 28; Figure 3) include the P38 mitogen-activated protein kinases (P38 MAPKs), which are responsive to many stressors and are involved in apoptosis, autophagy, and cell differentiation; AKT, also known as protein kinase B (PKB), important in signaling pathways regulating cell growth, proliferation, differentiation, and survival; IgG, a key protein activating the complement system for eliminating pathogens; and CAV1, a plasma membrane protein important in coupling integrins to the Ras-ERK pathway and promoting cell cycle progression. CAV1 also connects with a central class of histones (Histone H3) involved in the regulation of glucocorticoid signaling and other genes. The major themes of Network 2 are cellular morphology, cell-to-cell signaling and interaction, and hematopoiesis.

### 3.4. Gene Ontological Process Terms 

Analysis of gene ontological process terms enriched in the 200 differentially expressed genes revealed 12 terms that were statistically significant and potentially meaningful in relation to known effects of the TM program (Table 5).

### 3.5. Gene Classification Based on Associated Disease

Groupings of differentially expressed genes based on associated diseases are shown in Table 6. Sixty-two genes were related to hematologic diseases, 26 to coronary artery disease, 34 to diabetes complications, 49 to inflammation, and 64 to CVD. All these disease-related genes were downregulated in the TM group relative to the control group. 

### 3.6. Top Genes Upregulated in the Control Group, and Erythropoiesis-Related Genes

Table 7 shows the top six genes upregulated in the control group, along with four genes known to be related to erythropoiesis. As described in the Discussion, these top six genes are mainly involved in erythrocyte function.

## 4. Discussion

These results show first that the gene expression patterns obtained from the microarray analysis of small, demographically well-matched groups differ from each other in a manner consistent with expectations from prior research. Second, the qPCR results examining the relative expression of a sample of 15 key genes in these small groups as well as in larger, less well-matched groups appeared to uphold the trend of the microarray results. This was despite the poor demographic matching in the larger groups. Third, these expression differences appear to have functional confirmation from known stress effects on health and aging.

Several precautions concerning interpretation deserve mention. Because the group size for the validation step was relatively small, conclusions regarding possible TM effects cannot be generalized to larger populations without further confirming studies. A second precaution is that the control groups did not perform an activity that might qualify as a placebo for the twice-daily practice of TM programs for 38 years. Thirdly, it is possible that the two groups compared by microarray may have differed consistently in some unknown or poorly controlled variable.

The first important strength of the study is the finding of 200 genes in the discovery (microarray) component despite the use of comparatively strict inclusion criteria (expression ratio ≥ 2.0 and *p*-value ≤ 0.05). This ratio cut-off is quite high. Only 275 genes and loci (out of a total 16,247 in our study) reached the ratio criterion, while 2041 met the *p*-value criterion.

The second important strength is the existence of characteristic gene expression patterns for the two groups. The patterns are clear enough to be detected visually (Figure 1). More importantly, these patterns were derived objectively using hierarchical clustering analysis of the 200 genes, with data input consisting only of gene name, expression level, and participant signifier. The patterns derived are distinct and likely reflect functional differences rather than random differences in gene expression. These distinct patterns, combined with the random order of blood sampling and processing across both participants and days, argue against a transient stress or stimulus as the possible cause of between-group differences. 

The third and greatest strength lies in the fulfillment of predictions based on prior investigations. Many transcriptomic effects of chronic or extreme stress are known [23,24,25,45], and prior evidence exists for at least partial reversal of some of these effects by mind–body interventions [36], including by other techniques of meditation [32,33]. Furthermore, independent evidence exists showing that the TM program can reverse long-lasting effects of stress such as symptoms of PTSD [11,12,13], risk factors for CVD [15,16,17,18], chronically high levels of stress-related hormones [17,46,47], and low efficiency of energy metabolism [48]. Taken together, these prior studies predict that transcriptomic patterns associated with stress, such as the CTRA and low energy efficiency (see Introduction), should be prevented or reversed after long-term practice of this program. 

In the TM group, evidence of prevention or reversal of the CTRA can be seen first in the IPA network analysis of microarray data. Among the 14 genes in Network 1 (see Figure 2) that were downregulated in the TM group relative to the control group, 11 were associated with inflammation in the disease association analysis. On the other hand, among the 10 genes in Network 1 that were upregulated in the TM group, 7 were associated with the defense response in the analysis by gene ontological process term enrichment. 

Further indications that the TM group expresses a low-inflammation trait comes from the individual genes. The pro-inflammatory genes in Network 1 were either direct (e.g., *SOCS3*) or indirect (e.g., *ITGB3*) target genes of NF-κB. Expression of suppressor of cytokine signaling 3 (*SOCS3*) is known to correlate directly with pro-inflammatory cytokine levels [49], as are expression levels of integrin genes (*ITGB3*, *ITGB5*, and *ITGA2B)* [50]. A relative downregulation of these and 45 other genes related to inflammatory disease was found in the TM group, consistent with prevention or reversal of the primary, pro-inflammatory component of the CTRA. 

Based on the upregulation of genes related to disease resistance, e.g., those in the defense response and immune system processes categories from the analysis of gene ontological process term enrichment, seven of which also appear in Network 1, the TM group appeared to have enhanced antiviral, antibacterial, and anti-cancer activities, once again opposite to the CTRA pattern. The roles of specific differentially expressed genes further support this conclusion. Examples include five genes that are associated primarily with anti-cancer activity (*CXCL10*, *MICA*, *FPR2*, *CASP5*, and *CASP7)*, three genes that are associated with both anti-cancer and anti-microbial activity (*OAS1*, *ATF3*, and *IFIT3*), and four genes that are associated primarily with the defense response to viruses and bacteria (*CCL4L1*, *IL1B*, *ANKRD22*, and *TLR4*). This finding for genes upregulated in the TM group appears to confirm prevention or reversal of the second component of the CTRA.

This evidence for prevention or reversal of both components of the CTRA expression pattern raises another key point. Although initial evidence for the CTRA came from studies with severely stressed individuals [24,25], the results of the current study indicate a reduction in the CTRA in the TM group compared to healthy controls. This suggests that nominally healthy 65-year-olds carry a substantial load of stress effects, i.e., an allostatic overload [2,51], possibly due to accumulated effects of mild stressors. This is consistent with previous results from studies of short-term and long-term meditation practice indicating that the TM program reduces stress effects well below the level found in the general population [18,46,47].

Other evidence connecting stress effects with inflammation and disease is found in the relationship between genes grouped through ontological process term enrichment and genes grouped through disease association analysis. Using gene ontological term enrichment, 35 differentially expressed genes were classified as related to “response to stress.” Of these 35 genes, 27 also were found among the 49 genes classified under inflammation by disease association analysis, and 30 were found among the 64 genes classified under CVD, consistent with known associations between stress response, inflammation, and CVD. Furthermore, consistent with a close association between inflammation and CVD, 47 of the 49 inflammation genes also were found among the CVD-associated genes. Stress response genes were highly represented in the other disease categories as well. All these stress response genes, as well as all the disease-associated genes, were downregulated in the TM group relative to the control group, a direction more likely to be associated with benefits to health.

Another important prediction from prior research concerns evidence for a stress-induced reduction in energy efficiency. Recent articles by Picard, McEwen et al. summarize the critical roles that mitochondrial energy production and other mitochondrial functions play in stress and adaptation [27,31]. As reviewed by Jevning et al. [48], evidence that practice of TM programs increases energy efficiency includes decreased oxygen consumption, decreased respiratory rate, and decreased blood lactate levels. Lactate, a product of glycolysis that is produced in the blood mainly by erythrocytes, is increased during anaerobic metabolism and decreases acutely in erythrocytes during practice of TM [52]. Two observations in the present study provide evidence for higher energy efficiency in these long-term practitioners. Both may center on the role of SOCS3 in mitochondrial energy metabolism. 

*SOCS3*, downregulated in the TM group and prominent in Network 1, codes for a chemokine that is important in regulating energy metabolism through inhibitory effects on AMP-dependent protein kinase (AMPK) and leptin [53]. AMPK is central to energy metabolism in mitochondria, thus affecting cellular and whole-body energy levels [54,55]. Increased SOCS3 due to stress and increased inflammatory cytokines is documented to inhibit AMPK, causing insulin resistance in several tissues [56,57]. SOCS3 also can inhibit STAT3 activation [58], providing another possible route for decreased energy efficiency in mitochondria. Mitochondrial STAT3 plays a direct role in maintaining optimal function of the electron transport chain [59]. It is likely that removing inhibitory effects on AMPK by lowering SOCS3 contributes to the improved energy efficiency observed in practitioners of TM programs. 

The second observation is related to this and provides confirmatory evidence for more efficient energy production in the TM group. It involves hematologic system development and function, a major theme in the pathway analyses. The top six genes upregulated in the control group are mainly found in erythrocytes and are critical to erythrocyte function. Such large differences in expression of these genes likely reflect the presence of a greater number of reticulocytes (immature erythrocytes) in the control group blood samples. When erythrocyte production is high, reticulocytes, some of which may be large, even nucleated, can enter the bloodstream and contaminate PBMC samples. If mitochondrial oxidative phosphorylation is inefficient, more oxygen is required for a given level of energy production. Even moderate exercise, the level claimed by all study participants, is likely to produce chronic intermittent hypoxia in those with the lowest energy efficiency, and chronic intermittent hypoxia is known to increase erythropoiesis [60]. 

Supporting the hypothesis that inefficient mitochondrial energy production causes increased erythrocyte production in the control group, both *GATA1* and *GATA2*, master regulators of erythropoiesis [61], were significantly upregulated in this group. The increased expression of *GATA* normally causes an increase in erythropoietin, the direct stimulant of erythropoiesis [61]. Furthermore, *TAL1*, a regulator of erythropoietin receptor sensitivity [62], was significantly increased in the control group, and expression of *EPOR*, coding for the erythropoietin receptor, was increased as well, though not significantly. All these data are consistent with a substantially lower energy efficiency in the control group compared with the TM group. 

Another of the many potentially important observations in this study may deserve mention here. Based on the evidence that chronic stress can cause a decrease in telomere length (for review, see [63]), the increased *TAL1* expression found here in the control group is potentially a mediator of reduced telomerase activity. TAL1 inhibits the promotor of hTERT, the catalytic subunit of telomerase [64]. Elevated TAL1, therefore, could cause a reduction in telomerase activity and telomere length. 

Predictably, the larger, less well-matched groups of TM and control participants in which 15 key genes were studied by qPCR showed fewer statistically significant expression differences than were found in the microarray comparison of the well-matched groups. Nevertheless, for each of the main areas discussed, differential expression of one or more key genes reached significance. Thus, results for *SOCS3* and *ITGB5* verified an anti-inflammatory state; results for *SOCS3* and *AHSP* verified a state of enhanced energy efficiency, and the result for *CXCL10* (tumor suppressor) verified a higher defense response in the meditation group. It is expected that qPCR data on a larger sample of the 200 genes from the discovery step would give greater confirmation of these outcomes.

In an associated study in preparation [65], the transcriptomic data reported here were compared with cortisol and electroencephalographic (EEG) data from these larger groups, along with similar data from younger groups Results of that study tend to confirm the significance of the present findings in relation to proposed anti-stress and anti-aging effects of this meditation program.

## 5. Conclusions

The results of this study are intriguing and appear to provide strong directions for future studies. Finding 200 genes that met the dual cut-off criteria, and distinctive patterns of expression that appear to align with predictions from past research, provides a plausible framework. Many of the differentially expressed genes have previously demonstrated connections to stress effects, including to specific diseases related to stress. Although the possibility exists that the group differences are due to variables other than meditation practice, none of the data collected appear to shed light on this question. In addition to a future replication using RNA-seq, a larger *N*, and a larger sample of genes for validation, studies will be pursued on genes such as *SOCS3*, *AHSP*, and *CXCL10* that showed robust differences and have important functional implications.

## Figures and Tables

**Figure 1 medicina-57-00218-f001:**
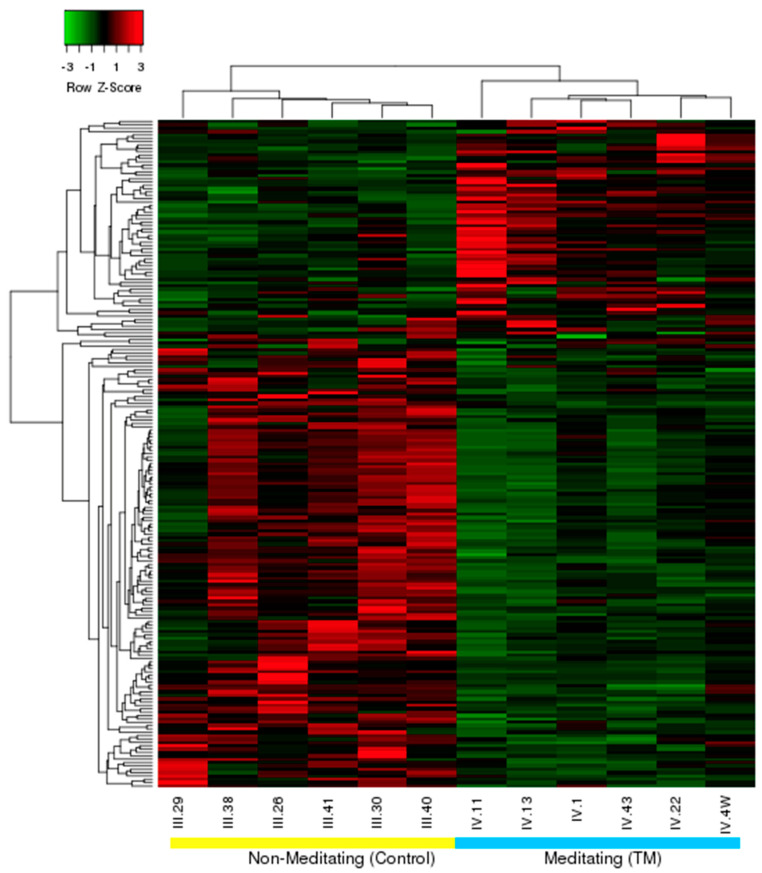
Heat map and hierarchical clustering of differentially expressed genes. The figure shows the relative expression of genes across participants, with hierarchical clustering of genes (rows) and participants (columns). Each colored bar indicates the degree to which the Z-score-normalized expression for that gene is either greater than (red) or less than (green) the median value for the gene.

**Figure 2 medicina-57-00218-f002:**
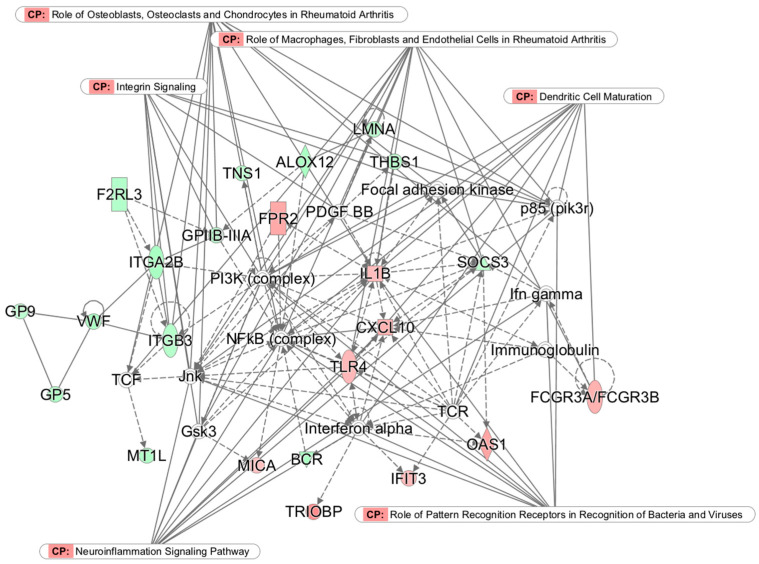
Network 1 (p-score 37): Cell-to-cell signaling and interaction, hematologic system development and function, and inflammatory response. Nodes without color denote non-significant genes, solid lines denote direct connections, and dotted lines denote indirect connections. Genes color-coded in red are upregulated and those in green are downregulated in the TM group. CP indicates canonical pathways.

**Figure 3 medicina-57-00218-f003:**
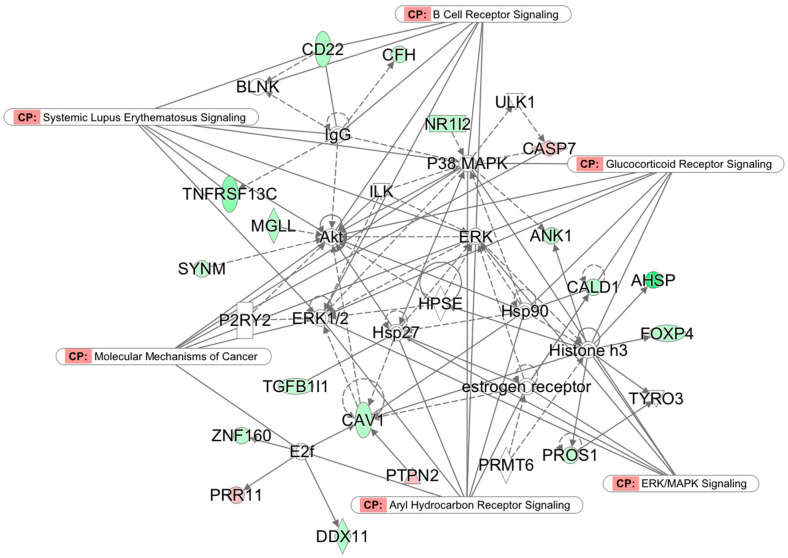
Network 2 (p-score 28): Cellular morphology, cell-to-cell signaling and interaction, and hematopoiesis. Nodes without color denote non-significant genes, solid lines denote direct connections, and dotted lines denote indirect connections. Genes color-coded in red are upregulated and those in green are downregulated in the TM group. CP indicates canonical pathways.

**Table 1 medicina-57-00218-t001:** Demographic Matching of Control and Transcendental Meditation^®^ (TM) Groups for Microarray Analysis.

Demographic Variables	Control Group	TM Group
*n*	6	6
Age (years ± SD)	65.0 ± 4.9	64.5 ± 5.4
Sex (number of males)	4	5
Non-vegetarians (number of)	5	5
Non-smokers (number of)	6	6
Non-drinkers (number of)	6	6
Moderate exercisers (number of)	6	6
Subjective SES * (mean)	3.0	3.1

* Subjective socioeconomic status (SES), using a 5-point scale from 1 for “lower class” to 5 for “upper class”.

**Table 2 medicina-57-00218-t002:** Demographic Matching of Control and TM Groups for qPCR Validation Analysis.

Demographic Variables	Control Group	TM Group
*n*	22	23
Age (years ± SD)	62.2 ± 4.64	63.6 ± 3.92
Sex (number of males)	10	14
Non-vegetarians (number of)	20	8
Non-smokers (number of)	17	23
Non-drinkers (number of)	17	21
Moderate exercisers (number of)	18	23
Subjective SES *	N/A	N/A

* Subjective SES data were not available for these participants.

**Table 3 medicina-57-00218-t003:** Comparison of Microarray Output with qPCR Output in the Small Groups.

Microarray Output (*N* = 12)			qPCR Output (*N* = 10 *)	
**Genes Downregulated in the TM Group Relative to the Control Group**				
**Gene**	***p*-Value**	**Expression Ratio ****	***p*-Value**	**“Fold Difference” *****
*AHSP*	0.002	5.45	<0.001	14.58
*ALOX12*	0.006	2.45	0.071	2.76
*CD22*	0.037	2.48	0.248	2.01
*ITGB3*	0.013	2.11	0.015	3.15
*ITGB5*	0.006	2.00	0.019	2.47
*LMNA*	0.004	2.16	0.063	2.35
*MYL9*	0.017	2.88	0.161	2.01
*SOCS3*	0.030	2.33	0.003	6.50
*TAL1*	0.002	2.36	0.073	2.19
**Genes Upregulated in the TM Group Relative to the Control Group**				
*CDKN1C*	0.017	2.79	0.003	2.28
*CXCL10*	0.014	2.89	0.084	2.33
*HES4*	0.047	2.00	0.748	1.12
*IGFBP7*	0.026	2.08	0.449	1.43
*IL1B*	0.012	3.54	0.126	1.77
*TLR4*	0.002	2.23	0.741	1.11

* TM participant 4 W lacked sufficient RNA for qPCR, so a demographically matched control (40) was dropped from the qPCR analysis, leaving *N* = 10. ** ratio of normalized mean expression values, reciprocal in the case of downregulation in TM. *** “Fold Difference” = 2 ^(∆∆Ct)^; Ct = threshold cycle.

**Table 4 medicina-57-00218-t004:** qPCR Determination of Relative Gene Expression (Ct) and “Fold Difference” * in Larger Groups.

Genes Downregulated in the TM Group Relative to the Control Group								
Gene	Group	Mean Ct	Std. Dev.	*N*	d*f*	F	*p*-Value	“Fold Difference” *
*AHSP*	Control TM	32.5668 34.5163	2.1561 1.4674	22 23	1	12.674	0.001	3.86
*ALOX12*	Control TM	30.6055 31.1267	0.7140 1.1756	22 23	1	3.194	0.081	1.44
*CD22*	Control TM	30.7102 31.0753	0.8499 1.1344	22 23	1	1.482	0.230	1.29
*ITGB3*	Control TM	26.4452 27.0511	0.7939 1.1830	22 23	1	4.031	0.051	1.52
*ITGB5*	Control TM	30.9789 31.4723	0.6233 0.7281	22 23	1	5.938	0.019	1.41
*LMNA*	Control TM	29.8050 30.2638	0.7014 0.9137	22 23	1	3.546	0.066	1.37
*MYL9*	Control TM	27.1693 27.5611	0.6926 1.1940	22 23	1	1.792	0.188	1.31
*SOCS3*	Control TM	31.4033 32.5206	0.9121 1.3372	22 23	1	10.626	0.002	2.17
*TAL1*	Control TM	30.7461 31.1823	0.6561 0.8969	22 23	1	3.441	0.070	1.35
**Genes Upregulated in the TM Group Relative to the Control Group**								
*CDKN1C*	Control TM	30.2448 29.9343	0.7156 0.7260	22 23	1	2.085	0.156	1.24
*CXCL10*	Control TM	32.7365 31.7125	0.9466 0.9613	22 23	1	12.950	0.001	2.03
*HES4*	Control TM	34.1132 34.3267	0.6820 0.7179	22 23	1	1.044	0.313	1.16
*IGFBP7*	Control TM	29.8499 29.5057	0.8188 0.8956	22 23	1	1.805	0.186	1.27
*IL1B*	Control TM	31.4125 31.0576	1.0132 0.8328	22 23	1	1.655	0.205	1.28
*TLR4*	Control TM	29.0808 28.8832	0.7732 0.7260	22 23	1	0.782	0.381	1.15

* “Fold Difference” = 2 ^(∆∆Ct)^; Ct = threshold cycle.

**Table 5 medicina-57-00218-t005:** Gene Ontological Process Terms.

**Term**	***p*-Value**	**Genes Downregulated in the TM Group**
Blood Coagulation	8.40 × 10^−8^	*F2RL3*, *CAV1*, *ITGB3*, *MMRN1*, *GP9*, *VWF*, *GP5*, *GP6*, *THBS1*, *TREML1*, *PROS1*, *ALOX12*, *ITGA2B*, *HBD*
Cell Activation	1.30 × 10^−4^	*F2RL3*, *CAV1*, *BCR*, *SNCA*, *TNFRSF13C*, *PAWR*, *ITGB3*, *GP9*, *VWF*, *GP5*, *GP6*, *THBS1*, *TREML1*, *WNT7A*, *ALOX12*, *ITGA2B*
Response to Stress	1.70 × 10^−4^	*SLC8A3*, *F2RL3*, *CAV1*, *CDC14B*, *SNCA*, *SLC6A4*, *FSTL1*, *PAWR*, *ITGB3*, *MMRN1*, *TRIM10*, *GP9*, *GP5*, *ALAS2*, *GP6*, *DDX11*, *PLOD2*, *CFH*, *MGLL*, *THBS1*, *HBD*, *BCR*, *PTPRF*, *SOCS3*, *LMNA*, *HBA2*, *HBA1*, *PTPRN*, *VWF*, *SH2D3C*, *TGFB1I1*, *TREML1*, *PROS1*, *ITGA2B*, *ALOX12*
Exocytosis	4.00 × 10^−4^	*VWF*, *ANK1*, *BCR*, *SNCA*, *SYTL4*, *ITGB3*, *THBS1*, *MMRN1*, *PROS1*, *ITGA2B*
Cell Adhesion	8.70 × 10^−4^	*CAV1*, *PTPRF*, *CALD1*, *TNFRSF13C*, *ITGB5*, *ITGB3*, *PAWR*, *MMRN1*, *GP9*, *KIFC3*, *VWF*, *GP5*, *CD22*, *CNTNAP2*, *SGCE*, *TGFB1I1*, *LAMC1*, *THBS1*, *JAM3*, *ALOX12*, *ITGA2B*
Hematopoiesis	1.30 × 10^−3^	*TAL1*, *ALAS2*, *AHSP*, *BCL11A*, *ZNF160*
**Term**	***p*-Value**	**Genes Upregulated in the TM Group**
Defense Response	2.40 × 10^−4^	*OAS1*, *CCL4L1*, *CXCL10*, *MICA*, *CASP5*, *FPR2*, *IFIT3*, *IL1B*, *LILRB2*, *METRNL*, *PTPN2*, *TLR4*, *VNN1*
Response to External Stimuli	3.30 × 10^−4^	*CCL4L1*, *CXCL10*, *MICA*, *ATF3*, *BATF3*, *CASP5*, *FPR2*, *IFIT3*, *IL1B*, *LILRB2*, *METRNL*, *KCNJ2*, *PTPN2*, *TLR4*
Inflammatory Response	9.30 × 10^−4^	*CCL4L1*, *CXCL10*, *FPR2*, *IL1B*, *METRNL*, *PTPN2*, *TLR4*, *VNN1*
Immune System Processes	1.20 × 10^−3^	*OAS1*, *CCL4L1*, *CXCL10*, *FCGR3B*, *MICA*, *FPR2*, *IFIT3*, *IL1B*, *LILRB2*, *PTPN2*, *TLR4*, *VNN1*
Homeostatic Processes	1.90 × 10^−3^	*CXCL10*, *CKB*, *FPR2*, *IL1B*, *METRNL*, *KCNJ2*, *PTPN2*, *SLC31A2*, *SLC4A8*, *TLR4*, *UTS2*
Cell Chemotaxis	2.50 × 10^−2^	GPR44, CCRL2, FPR2, CXCL10

**Table 6 medicina-57-00218-t006:** Differentially Expressed Genes Classified According to Associated Diseases (All Downregulated in the TM Group Relative to Control Group).

Disease	*p*-Value	Genes
**Hematologic Diseases**	1.5 × 10^−10^	*TRIM10*, *VPREB3*, *CFH*, *MGP*, *F2RL3*, *AHSP*, *TAL1*, *ITGB3*, *HBD*, *GATM*, *HRASLS*, *TMCC2*, *OSBP2*, *TNFRSF13C*, *PTCRA*, *DMTN*, *GP9*, *TNS1*, *CAV1*, *JAM3*, *CMTM5*, *BCL11A*, *SOCS3*, *SNCA*, *HBA2*, *SH2D3C*, *PTPRN*, *NR1I2*, *TGFB1I1*, *HDC*, *HBM*, *DDX11*, *PAWR*, *ITGB5*, *LMNA*, *ITGA2B*, *CABP5*, *THBS1*, *FOXP4*, *HBA1*, *ANK1*, *GP5*, *MAP1A*, *SLC35D3*, *CALD1*, *CD22*, *SLC4A1*, *BCR*, *LAMC1*, *ALOX12*, *HOMER2*, *TREML1*, *ALAS2*, *CA1*, *ABCB4*, *XK*, *EBF1*, *PTPRF*, *MMRN1*, *PLOD2*, *VWF*, *GP6*
**Coronary Artery Disease**	9.7 × 10^−8^	*CFH*, *MGP*, *F2RL3*, *ITGB3*, *MGLL*, *TNFRSF13C*, *CAV1*, *JAM3*, *SOCS3*, *PNOC*, *FSTL1*, *NDUFAF3*, *NR1I2*, *LMNA*, *ITGA2B*, *THBS1*, *HBA1*, *GP5*, *CALD1*, *PEAR1*, *EBF1*, *PTPRF*, *SLC6A4*, *VWF*, *GP6*, *FHL1*
**Diabetes Complications**	2.2 × 10^−7^	*CFH*, *MGP*, *F2RL3*, *ITGB3*, *GATM*, *SGCE*, *DMTN*, *CAV1*, *SOCS3*, *PNOC*, *FSTL1*, *HBA2*, *NDUFAF3*, *PTPRN*, *TGFB1I1*, *HDC*, *CNTNAP2*, *LMNA*, *ITGA2B*, *THBS1*, *SELENBP1*, *HBA1*, *ANK1*, *GP5*, *CALD1*, *CD22*, *SLC4A1*, *LAMC1*, *ALOX12*, *CA1*, *PTPRF*, *SLC6A4*, *VWF*, *GP6*
**Inflammation**	7.1 × 10^−6^	*CFH*, *MGP*, *CTDSPL*, *F2RL3*, *TAL1*, *ITGB3*, *MGLL*, *GATM*, *HRASLS*, *SLC8A3*, *TNFRSF13C*, *DMTN*, *WNT7A*, *GP9*, *CAV1*, *JAM3*, *SOCS3*, *SNCA*, *PNOC*, *FSTL1*, *HBA2*, *SH2D3C*, *PTPRN*, *NR1I2*, *HDC*, *PAWR*, *ITGB5*, *LMNA*, *ITGA2B*, *THBS1*, *SELENBP1*, *HBA1*, *ANK1*, *GP5*, *MAP1A*, *CALD1*, *CD22*, *SLC4A1*, *BCR*, *ALOX12*, *TREML1*, *ALAS2*, *CA1*, *ABCB4*, *EBF1*, *PTPRF*, *SLC6A4*, *MMRN1*, *VWF*
**Cardiovascular Disease**	4.4 × 10^−4^	*VPREB3*, *CFH*, *MGP*, *CTDSPL*, *F2RL3*, *AHSP*, *TAL1*, *ITGB3*, *HBD*, *MGLL*, *GATM*, *HRASLS*, *SGCE*, *SLC8A3*, *TNFRSF13C*, *DMTN*, *WNT7A*, *GP9*, *CAV1*, *JAM3*, *CMTM5*, *SOCS3*, *SNCA*, *PNOC*, *FSTL1*, *HBA2*, *NDUFAF3*, *PTPRN*, *NR1I2*, *TGFB1I1*, *HDC*, *DDX11*, *PAWR*, *CNTNAP2*, *ITGB5*, *LMNA*, *ITGA2B*, *THBS1*, *SELENBP1*, *FOXP4*, *HBA1*, *ANK1*, *GP5*, *MAP1A*, *CALD1*, *CD22*, *SLC4A1*, *BCR*, *LAMC1*, *ALOX12*, *PEAR1*, *ALAS2*, *CA1*, *ABCB4*, *XK*, *EBF1*, *PTPRF*, *SLC6A4*, *MMRN1*, *PLOD2*, *ZNF160*, *VWF*, *GP6*, *FHL1*

**Table 7 medicina-57-00218-t007:** Top Six Upregulated Genes in the Control Group and Upregulation of Genes Controlling Erythropoiesis.

Gene	Control Mean	Control SD	TM Mean	TM SD	Expression Ratio	*p*-Value
*HBM*	93.53	57.09	9.53	15.76	9.8	0.004
*SLC4A1*	72.47	49.02	8.13	10.54	8.9	0.002
*ALAS2*	631.43	579.80	78.12	67.67	8.1	0.002
*CA1*	85.37	77.98	12.40	9.03	6.9	0.002
*AHSP*	203.13	186.36	37.27	20.60	5.5	0.002
*HBD*	586.23	457.93	139.48	59.29	4.2	0.002
*GATA1*	19.22	8.31	11.17	4.27	1.7	0.049
*GATA2*	33.72	6.23	21.82	9.74	1.6	0.030
*TAL1*	64.28	9.70	27.18	6.22	2.4	0.002
*EPOR*	83.67	11.01	70.27	14.03	1.2	0.108

## Data Availability

Processed raw data from the microarray experiment have been deposited in the ArrayExpress database at EMBL-EBI and can be accessed at the following link: https://www.ebi.ac.uk/arrayexpress/experiments/E-MTAB-10252, accessed on 23 March 2021.

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
