# Peer review of "Transcriptomics of Long-Term Meditation Practice: Evidence for Prevention or Reversal of Stress Effects Harmful to Health"

_medicina, 2021, doi:10.3390/medicina57030218_

Round 1
Reviewer 1 Report
This submission presents the first transcriptomics investigation of a practise of meditation known as Transcendental Meditation (TM). It is of potentially of great importance since it can add to the data showing improved quality of life from the practise of TM and lead to its inclusion in general practice so people are not relying solely on pharmaceuticals for their health but on a combination of non-pharmaceutical approaches (meditation, chrononutrition, moderate exercise and so on).
However, there are number of issues that need to be addressed before publication can be considered.
There is a major omission in the statistical analysis of their transcriptomics data. The authors have considered p-values as if they were measuring the expression of only one gene. When you measure one feature it is perfectly fine to reject p-values, which are over 0.05 to keep an error rate of 5%. However, when you have more features (47,231 here) the probability of false discoveries increases a great deal. For example, you can consider that winning a game with an odd of 5% is a rare event if you play once and you win. However, if you play the same game 47,231 times and you have a probability of winning of 5%, you can expect to win more than 2,000 times only by chance. This is why it is essential to correct (or adjust) the p-values or provide the false discovery rate when one measures many features in ‘omics’ analyses. There is no indication that the authors have executed a correction or adjustment of p-values to take into account these multiple comparisons. In addition, most of their p-values are relatively high and thus in all likelihood they would not pass thresholds for significance after corrections by multiple comparisons. A correction or adjustment of p-values using for example the Benjamini-Hochberg Procedure needs to be undertaken to identify the number of genes whose level of expression is truly altered to a statistically significant degree.
It is stated that the data presented in Figure 1 is unsupervised. This is incorrect as only expression of the 200 genes they selected based on a supervised group comparison are shown. If the authors wish to do a true unsupervised clustering they should take the full dataset and show the levels of the 47,231 probes (or the gene expression levels for all genes). As this will be too much to be displayed as a heat map the authors need to reduce the dimension of the dataset by a principal component analysis (PCA) method.
The validation of gene expression by qPCR and that these values are concordant with the microarray data is reassuring. However, as the authors did not correct for multiple comparisons in this analysis as mentioned above, and some p-values are relatively high, it is not possible to draw any definitive conclusions from the qPCR results. The qPCR by itself is sufficient to say that the expression of these genes may be changed (it is not fully sure even with these p-values) but it is not sufficient to corroborate the conclusions on functional changes. Having very low p-values in network analyses is not surprising even when there is no effect because the starting point is biased.
The authors provide an explanation for the possible number of random occurrence in gene expression changes in the discussion section saying that 35 genes can be expected to vary by chance but they don’t provide explanations on how they conducted this calculation and arrived at this figure. This also does not change the fact that they should have corrected their p-values.
Overall, the study suffers from being highly statistically underpowered. It does not mean that the intervention did not have an effect, it means that their design may not be sufficient to detect it. Only after p-value correction will it be possible to see clearly what can and cannot be deduced from the dataset as presented.
Finally, at best this investigation can only be considered as a pilot study and should be presented as such. The current conclusions not corroborated by the data analysis, which needs major improvement.
Author Response
Re: Manuscript ID: medicina-1042090
Type of manuscript: Article
Title: Transcriptomics of long-term meditation practice: Evidence for
prevention or reversal of stress-effects harmful to health
Authors: Supaya Wenuganen *, Kenneth G Walton *, Shilpa Katta, Clifton L
Dalgard, Gauthaman Sukumar, Joshua Starr, Frederick T Travis, Robert Keith
Wallace, Paul Morehead, Nancy K Lonsdorf, Meera Srivastava *, John Fagan *
Received: 1 December 2020
Reviewer 1: We appreciate your careful review of our study. We especially appreciate your pointing out methodological details that need to be included for readers that may not be current on state-of-the-art bioinformatics and high-throughput gene expression methodologies. Here are our point-by-point replies to the questions raised in your review.
- Did statistical analysis of microarray data adjust for errors, especially Type 1 (false positive) errors?
Response: The short answer is “yes,” but we failed to include information in our original submission that would have made this more obvious. We also failed to present a broader perspective on the role of microarray analysis in achieving the overall goal of identifying and understanding differentially expressed (DE) genes. We have now clarified the point that statistical analysis using R BioConductor (lumi and BeadArray) packages includes corrections for large numbers of genes (see lines 135 and 138 in the text). We also have inserted a short statement in the Introduction (see lines 77-82) that outlines more clearly the role of microarray in studying DE between two sets of samples, with an appropriate reference (see https://biologydirect.biomedcentral.com/articles/10.1186/s13062-015-0077-2 ).
Microarray analysis is an important first step, also called the discovery step, in identifying potentially significant genes. The second step is to more precisely assess these genes using more accurate methods such as real-time PCR. However, rtPCR is not practical for large scale analyses, e.g. more than one or two dozen genes, due to the laborious steps involved. Therefore, the microarray analysis comparison of thousands of genes serves an important “weeding out” function. The third step in comparing gene expression across groups is evaluation of the meaningfulness of specific genes and their interactions. This depends on the biological function of the genes and whether the effects make sense in relation to prior studies or to results of planned subsequent biological/biochemical studies.
All steps in microarray analysis have been researched for accuracy, and the best performing statistical approaches are included in the analysis software packages. That includes the first stage called annotation, that is, inferring the genes and loci based on multiples of oligonucleotide probes present in the array. In our study, 47,231 probes or features led to identification of 16,247 genes and loci. Of these genes and loci, 2041 fell within our p ≤ 0.05 cutoff for DE genes. However, it is important to note that we chose the second cutoff variable, the ratio of relative rates (≥ 2.0), to be quite stringent. This step alone reduced the total possible DE genes to only 275. Combining the two criteria led to only 200 genes. It is highly likely that many more of the genes are DE, but we chose in this first study to use stringent criteria to further reduce potential false positives. These facts are now made clearer in the text (see lines 278-281).
- Was the term “unsupervised hierarchical clustering” used inappropriately to describe the clustering aspect of the heat map of DE genes (Fig 1)?
Response: Clustering analysis exists in many types and is used for multiple purposes. In the present case, the hierarchical clustering analysis method used in conjunction with the heat map is a common one used to discern distinct patterns of gene expression. With the algorithm used here to detect different patterns of expression by the two groups, the only data input were gene name, expression level, and participant signifier. This satisfies a narrower definition of “unsupervised” for this set of numbers. However, as the reviewer points out, the reason participants aggregated into two groups (TM and control) was because the 200 genes had already been selected due to the criteria for differential expression. We also appreciate the reviewer’s mention of principal components analysis. Such a method for reducing variables would be an alternative to the high ratio of expression rates (≥ 2.0) that we used and will be considered for future studies. Nevertheless, the different patterns seen to characterize gene expression in our two groups may argue for systematic differences related to function rather than to random differences in expression. Changes were made in the text to clarify these points (see lines 282-287).
- Interpretation of the qPCR data as definitive validation of the microarray results or of the likely functional differences discussed.
Response: As the second step in comparison of gene expression differences, qPCR adds another level of quantitative evaluation. As the reviewer points out, the correspondence between the parameters measured in the microarray analysis and those in the qPCR was close, but not identical, for the small groups. This discrepancy could have been due to real differences between the microarray and PCR results or it could have been due to the unfortunate loss of one sample (due to not meeting the qualifying criterion for RNA) and the necessity of dropping the closest matching sample from the other group to maintain equal numbers in the groups. Considering this change in the sample sets, the correspondences for most of the genes seem surprisingly good. The next step, an attempt to validate these results in larger groups of study participants using qPCR, was hampered by the demographic match not being as close as for the microarray component and by the limited number of genes (15) that we could afford to examine. The reviewer is correct in saying these data are not sufficient to draw definitive conclusions concerning the functional differences we discussed. For a definitive study, one would need larger groups. Ideally a greater number of genes also would be compared in this second step. A note to this effect is inserted in the text (see lines 35, 393-394, 400-408).
Nevertheless, the functional interpretation of results is considered the vital third step for understanding gene expression differences. Whether from previous transcriptomic and non-transcriptomic research on stress and meditation or from future studies chosen to test the validity of biological effects, interpretation of the transcriptomic results must be understandable and supported by the functional roles of DE genes. The fact that DE differences apparent in this study provide clear links to previous functional results from prospective, randomized, controlled trials of this and other forms of meditation may be the strongest confirmation of the importance of the observed transcriptional differences. To date, over 400 peer reviewed studies of effects of short-term Transcendental Meditation practice have identified reduction of stress effects as perhaps the major biologically observable function of this type of meditation. Reduction of stress effects alone may underlie benefits observed in the areas of PTSD, anxiety and depression, cardiovascular disease, energy efficiency etc. (See Introduction and Discussion for references to some key studies.) As for future studies, the present results point to important directions to pursue in larger studies, especially the strong links to the CTRA and energy efficiency, both of which relate in major ways to the robust difference in SOCS3 expression. Prior to this study, we were not aware of the potential role of this gene in mediating several downstream effects of stress nor of its potential as an indicator of effects of this meditation technique. Among the 15 genes selected for qPCR (which were by no means the only ones we might have selected) there are at least 2 others (AHSP, CXCL10) whose robust DE make them strong candidates for specific follow-up.
- Having very low p-values in network analyses is not surprising even when there is no effect because the starting point is biased.
Response: We are unclear what is meant by “the starting point is biased.” If it is meant to imply investigator bias, this is probably not valid. The only investigator input was the relative expression levels of the 200 genes and the choice of one of four causal analytic tools in the Ingenuity Pathway Analysis software package. The person conducting this analysis was unfamiliar with prior research on the TM techniques. The networks shown are those that emerged when the 200 were entered. Those with the highest p scores were selected for publication. Although there appears to be no consensus in the literature concerning the minimum p score for a network to be taken seriously, certainly the two shown in Results qualify as worthy of follow-up. (If this does not properly address the reviewer’s point, further clarification by the reviewer is necessary.)
- Did not provide explanation for how random occurrences could account for 35 DE genes.
Response: This was simply a calculation of the expected number of genes to meet both criteria based on the product of the observed frequencies with which each of the individual cutoff criteria were met, i.e. (2041/16,247 x 275/16,247) x 16,247 = 34.55. This assumes that the observed frequencies of meeting the individual criteria were due to chance alone. This text is now modified to increase clarity (see lines 278-281). Few readers would expect chance alone to be responsible for the observed outcomes.
- Statistically underpowered. This investigation can be considered only as a pilot study.
Response: The intention of this first study was clearly exploratory in nature. However, as explained in response to the reviewer’s first question and in the brief statement added to introduction (lines 77-82), microarray analysis is the first of three main steps for reaching conclusions about transcriptomic differences. It must be followed by a second step, i.e., a more quantitative method such as qPCR. The third step is evidence supporting the functional significance and relevance for the comparison in question. Functional evidence can be from previous related studies, or it can be new evidence generated from new studies. The microarray analysis casts the broad net. It can be made stricter by using multiple criteria, as in our study. The number of samples per condition is often small, 5 or fewer, due to costs (see https://www.ebi.ac.uk/training-beta/online/courses/functional-genomics-ii-common-technologies-and-data-analysis-methods/microarrays/analysis-of-microarray-data/differential-expression-analysis/).
In lines 77-78, the point is now made that this is the initial study of its type and is exploratory in nature.

Reviewer 2 Report
This is a well-written small study on an important topic, understanding the physiological basis for the health benefits of transcendental meditation. The authors identify gene expression features in PBMCs that differ between long-term meditators and matched controls. The results are interpreted in terms of pathways for which there was previous evidence for an impact of practicing TM. PBMCs are a natural cell type to examine for the pathways that emerged as behaving differently in the two groups: energy efficiency, inflammation, and immune function.
The main limitation of the study, which is otherwise well done, is that the discovery cohort used to identify differential gene expression between the TM and control groups consisted of n=6 in each group. Moving more of the subjects from the validation to the discovery cohorts while maintaining the matching between TM and control groups could have revealed additional differentially expressed genes and reduced the influence of individual variation within the groups. Future studies should use larger discovery groups and employ RNA-seq instead of bead arrays to cast a wider and more unbiased net for differentially expressed genes. However, the fact that the markers that emerged from the array analysis could be validated in a larger cohort of TM and control subjects is encouraging. It would be a plus for the authors to comment on next steps for follow-on studies as well as the reason why so few subjects were used in the discovery cohorts (perhaps cost?).
Other points that should be addressed:
- Lines 187-188 and Fig. 1: What was up-regulation and down-regulation relative to, since both the TM and control groups show changes?
- Table 3: define df and F; how was “validation done” on genes that were not represented in Table 2 (array)? How were those genes chosen?
- 2: define non-colored nodes and dotted vs. solid lines connecting nodes.
- Table 4 and lines 236-237: Were there terms that were statistically significant and yet not interpretable in terms of known effects of TM?
- In the validation analysis, did any effect emerge of the parameters listed in Table 1, which are potential confounders, particularly sex? (In the discovery groups only 1-2 subjects were female.) A table similar to Table 1 should be provided for the validation group, which is described as less well-matched in terms of potential confounding variables.
- The authors should consider beefing up the Results section and shortening the Discussion. Much of the commentary about the genes highlighted in Tables 4-6 and the networks could be presented in the Results section, which is currently rather thin, while leaving the interpretation to the Discussion.
Author Response
Re: Manuscript ID: medicina-1042090
Type of manuscript: Article
Title: Transcriptomics of long-term meditation practice: Evidence for
prevention or reversal of stress-effects harmful to health
Authors: Supaya Wenuganen *, Kenneth G Walton *, Shilpa Katta, Clifton L
Dalgard, Gauthaman Sukumar, Joshua Starr, Frederick T Travis, Robert Keith
Wallace, Paul Morehead, Nancy K Lonsdorf, Meera Srivastava *, John Fagan *
Received: 1 December 2020
Reviewer 2: Your review and questions are helpful and are much appreciated. Here are our point-by-point replies.
- The n for discovery groups is small. Why did we not move more subjects from the validation group into the discovery groups?
Response: Our goal was to have as many well-matched participants as our budget could afford in the TM and Control groups for the discovery (microarray) step. However, we exhausted the list of well-matched participants at 7 x 7. After extraction, one sample was found to have RNA that did not meet our criteria for purity, necessitating dropping another participant to maintain equal sized groups.
- Future studies should use larger discovery groups and RNA-seq instead of bead arrays to cast a wider net for DE genes.
Response: Our intention is to replicate this study using somewhat larger groups, both at the discovery step and at the validation step. The plan is to use RNA-seq in the discovery step. This goal is now mentioned in the text. (See answer to number 3.)
- Authors should comment on next steps for follow-on studies and why so few were used in the discovery cohort (perhaps cost?)
Response: (See answers to questions 1 and 2.) Aside from a larger replication using RNA-seq in the discovery phase, we plan to investigate more thoroughly the possible roles of genes related to the main apparent effects from this first study, namely, anti-inflammatory genes, defense response genes, and genes related to energy efficiency. These would include especially those that showed the most robust expression differences in this exploratory study (SOCS3, AHSP, and CXCL10). A statement on follow-on studies was inserted (lines 407-409). Also, as alluded to in the original version of the manuscript, we have data related to anti-aging effects that were obtained from this study, namely data on hair cortisol concentration, an EEG measure reflecting cognitive processing speed, and transcriptomic data (qPCR for the same 15 genes studied here but in a young non-TM group). A version of the present paper that included those data was deemed too long to be submitted as one paper, leading to the decision to publish those data in a second paper.
- (Reviewer’s Other Point 1) Lines 187-188 and Fig. 1: What was up-regulation and down-regulation relative to, since both the TM and control groups show changes?
Response: Expression values for each gene were Z-score normalized across all participants. The illustrated colors then represent the degree to which a specific expression value was larger (red) or smaller (green) than the median value. Changes have been entered in the figure legend to make this clear (lines 193-197).
- (Reviewer’s Other Point 2) Table 3: define df and F; how was “validation done” on genes that were not represented in Table 2 (array)? How were those genes chosen?
Response: The degrees of freedom (df) and the F-statistic came from the statistical comparison of the two groups. Tables 2 and 3 list the same genes. These were chosen somewhat arbitrarily, but all were deemed important to proposed anti-stress or antiaging effects of TM. We have now made clear (see lines 77-83, 394, 395, 407-409) that the results of qPCR analysis for this sample of 15 genes add a degree of confidence that other DE genes from the microarray can be taken as potentially meaningful if previous or future information on their function fits with previously known or newly discovered biological effects of the meditation practice.
- (Reviewer’s Other Point 3) For network charts, define non-colored nodes and dotted vs. solid lines connecting nodes.
Response: The following statement is now added to the figure legend for networks. “Nodes without color denote non-significant genes, solid lines denote direct connections and dotted lines denote indirect connections.”
- (Reviewer’s Other Point 4) Table 4 and lines 236-237: Were there ontological process terms that were statistically significant and yet not interpretable in terms of known effects of TM?
Response: The ontological process terms listed in (old) Table 4 were those with highest statistical significance and were listed regardless of whether we had prior evidence of effects of TM on these processes.
- (Reviewer’s Other Point 5) In the validation analysis, did any effect emerge of the parameters listed in Table 1, which are potential confounders, particularly sex? (In the discovery groups only 1-2 subjects were female.) A table similar to Table 1 should be provided for the validation group, which is described as less well-matched in terms of potential confounding variables.
Response: A new table is now included (see lines 113-114) and mention of possible confounders is made on lines 405-407. Although the TM group contained more participants reporting a vegetarian diet, this was not the case among the participants in the discovery (microarray) component.
- (Reviewer’s Other Point 6) The authors should consider beefing up the Results section and shortening the Discussion. Much of the commentary about the genes highlighted in Tables 4-6 and the networks could be presented in the Results section, (which is currently rather thin, while leaving the interpretation to the Discussion.
Response: The reviewer’s point is well taken, but we feel changes of this magnitude cannot be completed adequately in the short time allowed for returning the manuscript. If more time were available, such modifications could be undertaken.

Reviewer 3 Report
The authors must appreciate that sample size calculation is an important aspect for conducting such a study. Ideally, samples should not be too small. This may prevent the findings from being extrapolated.
Author Response
Re: Manuscript ID: medicina-1042090
Type of manuscript: Article
Title: Transcriptomics of long-term meditation practice: Evidence for
prevention or reversal of stress-effects harmful to health
Authors: Supaya Wenuganen *, Kenneth G Walton *, Shilpa Katta, Clifton L
Dalgard, Gauthaman Sukumar, Joshua Starr, Frederick T Travis, Robert Keith
Wallace, Paul Morehead, Nancy K Lonsdorf, Meera Srivastava *, John Fagan *
Received: 1 December 2020
Reviewer 3: Your review is much appreciated. Here is our response to your comment that the sample size is too small and prevents the findings from being extrapolated (presumably, to larger populations).
Response: This is the first study of its kind on long-term practitioners of TM. Because we did not have a preliminary study, we could not calculate the sample size. The intention of this first study was clearly exploratory in nature. However, we understand that a microarray analysis is the first of three main steps for reaching conclusions about transcriptomic differences. It must be followed by a second step, i.e., a more quantitative method such as qPCR. The third step is evidence supporting the functional significance and relevance for the comparison in question. Functional evidence can be from previous related studies, or it can be new evidence generated from new studies as a later step. The microarray analysis casts the broad net. It can be made stricter by using multiple criteria, as in our study. The number of samples per condition is often small, 5 or fewer, due to costs and other factors (see
https://www.ebi.ac.uk/training-beta/online/courses/functional-genomics-ii-common-technologies-and-data-analysis-methods/microarrays/analysis-of-microarray-data/differential-expression-analysis/).
Our goal was to have as many well-matched participants as our budget could afford in the TM and Control groups for the discovery (microarray) step. However, we exhausted the list of well-matched participants at 7 x 7. After extraction, one sample was found to have RNA that did not meet our criteria for purity, necessitating dropping another participant to maintain equal sized groups. In the verification and validation “second step” a subset of genes from the microarray was studied more accurately using quantitative PCR with 22 Control vs 23 TM participants. In general, although these subjects were less well matched demographically, the direction of changes was retained and several genes showed robust differences in expression that matched those found in the discovery step.

Round 2
Reviewer 1 Report
The authors have satisfactorily addressed most of the concerns I raised at the time of first review.
The main point I raised regarding the statistical analysis used has been clarified. The authors state that they used "R BioConductor (lumi and BeadArray) packages includes corrections for large numbers of genes". In this light the authors should provide a table with the p-values and the adjusted p-values for the 200 genes whose expression they purport to be altered to a significant degree. On the software webpage (https://www.bioconductor.org/packages//2.7/bioc/vignettes/lumi/inst/doc/lumi.pdf), under point 9.2. called 'identify differentially expressed genes', there is an example output of a table showing the different genes with their p-values and adjusted p-values (page 36). The authors should show the same for the 200 genes whose expression is shown to be altered by their microarray analysis and thus provide convincing evidence for statistically significant differences. This data can be shown as supplementary online material to allow readers to peruse this data.
Once this data is made available and the main text adjusted with due reference to the table then I would recommend publication.